# Reliable Active Learning from Unreliable Labels via Neural Collapse Geometry

**Atharv Goel**[*]
IIIT Delhi
atharv21027@iiitd.ac.in

**Sharat Agarwal**[*]
IIIT Delhi
sharata@iiitd.ac.in

**Saket Anand**
IIIT Delhi
anands@iiitd.ac.in

**Chetan Arora**
IIT Delhi
chetan@cse.iitd.ac.in

## Abstract

Active Learning (AL) promises to reduce annotation cost by prioritizing informative samples, yet its reliability is undermined when labels are noisy or when the data distribution shifts. In practice, annotators make mistakes, rare categories are ambiguous, and conventional AL heuristics (uncertainty, diversity) often amplify such errors by repeatedly selecting mislabeled or redundant samples. We propose Reliable Active Learning via Neural Collapse Geometry (NCAL-R), a framework that leverages the emergent geometric regularities of deep networks to counteract unreliable supervision. Our method introduces two complementary signals: (i) a **Class-Mean Alignment Perturbation score**, which quantifies how candidate samples structurally stabilize or distort inter-class geometry, and (ii) a **Feature Fluctuation score**, which captures temporal instability of representations across training checkpoints. By combining these signals, NCAL-R prioritizes samples that both preserve class separation and highlight ambiguous regions mitigating the effect of noisy or redundant labels. Experiments on ImageNet-100 and CIFAR100 show that NCAL-R consistently outperforms standard AL baselines, achieving higher accuracy with fewer labels, improved robustness under synthetic label noise, and stronger generalization to out-of-distribution data. These results suggest that incorporating geometric reliability criteria into acquisition decisions can make Active Learning less brittle to annotation errors and distribution shifts, a key step toward trustworthy deployment in real-world labeling pipelines.

## 1 Introduction

Deep learning depends on large-scale annotations (6), but real-world labels are often unreliable. This undermines Active Learning (AL) (9), whose heuristics (uncertainty (11; 4), diversity (9; 1)) can even exacerbate noise by selecting mislabeled, redundant, or ambiguous samples. This leads to inefficient label use, degraded generalization, and poor robustness under distribution shifts (3; 10).

Neural Collapse (NC) theory (7) shows that, late in training, features concentrate near class means, which align as a simplex ETF. These regularities provide stability even under imperfect supervision, suggesting that sample selection guided by NC dynamics could improve both efficiency and robustness (5; 2).

In this paper, we propose NCAL-R, a Neural Collapse–guided Active Learning framework designed for reliability under noisy or uncertain supervision. By quantifying how candidate samples perturb

---

[*]Equal contribution.

inter-class alignment and fluctuate across training checkpoints, NCAL-R selects points that both preserve feature structure and expose genuine ambiguities. Our experiments demonstrate improved accuracy with fewer labels, enhanced robustness to synthetic noise, and stronger out-of-distribution generalization.

## 2 Methodology

**Problem Setting.** We consider a standard pool-based Active Learning (AL) setting: a small labeled set $\mathcal{L}$, a large unlabeled set $\mathcal{U}$, and a model $f_\theta$ trained on $\mathcal{L}$. At each acquisition step, an AL strategy selects a batch $\mathcal{B} \subset \mathcal{U}$ for annotation. Our goal is to select $\mathcal{B}$ such that the learned representation is *robust* to both covariate shift and label drift, enabling improved in-distribution accuracy, OOD detection, and novel-class discovery.

**Neural Collapse as a Structural Signal.** In the late phase of training, deep classifiers often exhibit *Neural Collapse* (NC) (7): (NC1) within-class feature variance collapses, (NC2) class means form vertices of a simplex equiangular tight frame (ETF), (NC3) classifier weights align with class means, and (NC4) classification reduces to nearest-class-mean decisions. This emergent geometry reflects high class separability; deviations from it, or instability within it, may indicate *structurally valuable* samples that, when labeled, can improve generalization.

**Acquisition Metrics.** NCAL-R computes two complementary scores for each $x \in \mathcal{U}$:

1. **Class-Mean Alignment Perturbation (CMAP)**: Let $\mu_c$ denote the current empirical mean feature vector of class $c$ and let $\hat{y}(x)$ be the model's predicted class for $x$. Denote by $z$ the penultimate-layer feature for $x$. For any vector $h$ we write $\bar{h} := h/\|h\|$ for its $\ell_2$-normalized version. The class-mean updated by including $z$ in class $c$ (which has $n_c$ members) is

$$\tilde{\mu}_c = \frac{n_c \mu_c + z}{n_c + 1}.$$

   Define the sum of normalized class means by $\bar{M} := \sum_{i=1}^{C} \bar{\mu}_i$, where $C$ is the number of classes. We quantify the perturbation induced by $x$ as the change in alignment of the (normalized) class mean with respect to the average of other class means:

$$\boxed{\text{CMAP}(x) := \left(\bar{\tilde{\mu}}_c - \bar{\mu}_c\right)^\top \left(\bar{M} - \bar{\mu}_c\right),} \tag{1}$$

   where $c = \hat{y}(x)$. Intuitively, $\delta_x$ measures how much adding $x$ shifts its predicted-class mean toward (or away from) the centroid of the other class means; large positive values indicate samples that significantly perturb inter-class geometry and are therefore likely to refine decision boundaries. We derive this result in the Appendix.

2. **Feature Fluctuation (FF)**: Given model checkpoints $\{\theta_t\}_{t=T_i}^{T_f}$ where $T_i$ and $T_f$ are the start and end epochs in the NC phase, let $s_{\theta_t}(x) \in \mathbb{R}^c$ denote the pre-softmax logit vector produced for sample $x$. FF measures the variance of predicted logits for $x$ across $\theta_t$. High FF identifies samples with persistent uncertainty, even when most features have stabilized.

$$\boxed{\text{FF}(x) = \sum_{t=T_i+1}^{T_f} \mathbf{1}\left[\arg\max s_{\theta_t}(x) \neq \arg\max s_{\theta_{t-1}}(x)\right]} \tag{2}$$

**Combined Acquisition Strategy.** NCAL-R selects the top-$k$ samples from $\mathcal{U}$ by ranking CMAP and FF separately, standardizing each by their mean and standard deviation, and averaging:

$$\text{Score}(x) = \frac{\text{CMAP}(x) + \text{FF}(x)}{2}.$$

This yields a batch $\mathcal{B}$ that contains both structurally impactful and prediction-unstable samples, shaping the representation to be both discriminative and adaptable. NCAL-R requires no auxiliary networks, pseudo-labeling, or task-specific tuning, and can be applied to any backbone or modality where feature embeddings can be extracted.

# 3  Experiments

**Experimental Setup.**   We evaluate NCAL-R on tasks including classification, OOD detection, OOD generalization, and general category discovery. Label drift is tested under the GCD protocol; covariate shift via linear probes on OOD datasets. Unless noted, we use a ResNet-18 backbone, 5% acquisition per cycle, and compare to Random, CoreSet (9), and CDAL (1).

**Evaluation Metrics.**   We report: (i) **All-class accuracy**: top-1 classification accuracy over both known and novel classes; (ii) **Novel-class accuracy**: GCD accuracy restricted to novel classes; (iii) **Known-class accuracy**: classification accuracy on known classes; (iv) **AUROC** for binary OOD detection between in-distribution and OOD samples.

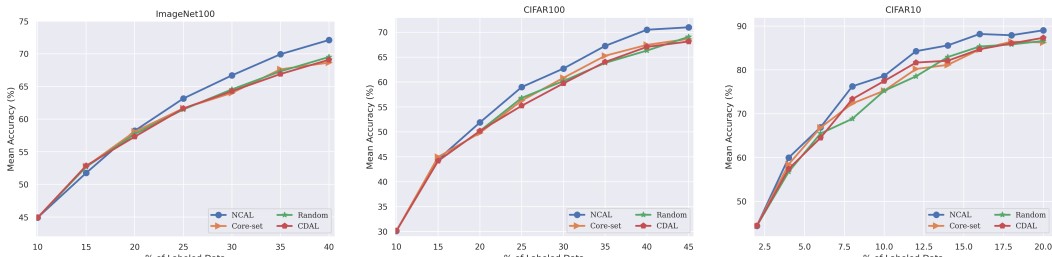

Figure 1: Comparison of test accuracy across varying label budgets on three benchmark datasets—ImageNet100, CIFAR100, and CIFAR10. NCAL's good performance even at lower annotation budgets suggests that its Neural Collapse-guided selection promotes more structured and representative feature learning. *(Note: accuracy for 100% data of ImageNet100, CIFAR100 and CIFAR10 are: 79.16%, 70.75% and 90% respectively. Reported results are average of 3 independent runs.)*

| Method | 10% | 15% | 20% | 25% | 30% | 35% | 40% | 100% |
|---|---|---|---|---|---|---|---|---|
| Random | | 80.57 | 84.13 | 85.45 | 86.89 | 87.82 | 88.67 | |
| CDAL | 77.18 | 81.78 | 84.28 | 85.9 | 86.34 | 87.98 | 88.92 | 93.68 |
| Coreset | | 81.56 | 83.73 | 85.66 | 87.1 | 88.29 | 88.95 | |
| NCAL | | **82.49** | **85.55** | **87.89** | **89.15** | **90.53** | **91.53** | |

Table 1: AUROC scores for Far-OOD detection on the OpenImage-O dataset trained on ImageNet-100 with varying annotation budgets.

| Method | All Classes | Old Classes | New Classes | Val Accuracy |
|---|---|---|---|---|
| Random | 33.20 | 50.34 | 20.35 | 36.20 |
| CDAL | 33.39 | 49.96 | 20.96 | 36.94 |
| Coreset | 32.23 | 49.98 | 18.92 | 36.44 |
| **NCAL** | **35.07** | **51.95** | **23.05** | **37.76** |

Table 2: Performance across all, old, and new classes along with validation accuracy.

**Covariate Shift Results.**   We test the ability to generalize to OOD datasets by training a linear probe over the learned embeddings. Table 4 shows that NCAL-R improves OOD classification by $\sim 2\%$ on average across 8 varying datasets, over all baselines. This demonstrates the adaptability of NCAL-R's feature space to both NearOOD and FarOOD scenarios.

**Label Drift and GCD.**   NCAL-R's geometry-aware selection yields features that support unsupervised novel-class discovery while maintaining high accuracy on known classes. In the GCD setting with 60-40 split, NCAL-R improves novel-class accuracy by $+2.1$ points over the best baseline without supervision on novel classes, and by $+1.6$ points on known classes. This demonstrates that NCAL-R's feature space is inherently adaptable to evolving label spaces, without forgetting past label information.

| Method | 10% | 15% | 20% | 25% | 30% | 35% | 40% | 100% |
|---|---|---|---|---|---|---|---|---|
| Random | | 80.57 | 84.13 | 85.45 | 86.89 | 87.82 | 88.67 | |
| CDAL | 77.18 | 81.78 | 84.28 | 85.9 | 86.34 | 87.98 | 88.92 | 93.68 |
| Coreset | | 81.56 | 83.73 | 85.66 | 87.10 | 88.29 | 88.95 | |
| NCAL | | **82.49** | **85.55** | **87.89** | **89.15** | **90.53** | **91.53** | |

Table 3: AUROC scores for Far-OOD detection on the OpenImage-O dataset trained on Imagenet-100 with varying annotation budgets.

| | Val / Train Acc | | OOD Generalization (linear probe val accuracy) | | | | | | | | |
|---|---|---|---|---|---|---|---|---|---|---|---|
| | Val (%) | Train | ImgNet-R | CIFAR100 | Flowers | NINCO | CUB | Aircraft | Pets | STL | Avg |
| Random | 69.51 | 96.44 | 18.06 | 41.64 | 58.69 | 64.23 | 37.84 | 15.26 | 42.34 | 68.67 | 46.95 |
| CDAL | 69.09 | 96.55 | 17.56 | 41.98 | 58.13 | 65.87 | 38.53 | 15.03 | 42.65 | 68.27 | 47.21 |
| Coreset | 68.65 | 96.42 | 16.93 | 42.02 | 57.96 | 65.11 | 37.86 | 15.15 | 42.22 | 68.68 | 47.00 |
| NCAL | **72.11** | **95.22** | **19.27** | **43.78** | **60.87** | **67.66** | **40.01** | **15.38** | **44.70** | **70.49** | **48.98** |
| 100% | 79.16 | 95.27 | 20.01 | 45.31 | 61.77 | 69.90 | 42.29 | 19.08 | 46.14 | 71.45 | 50.87 |

Table 4: Comparison of validation accuracy, Neural Collapse metrics, and OOD generalization (measured via linear probe accuracy) across multiple benchmarks. NCAL consistently achieves stronger generalization to diverse OOD datasets compared to baselines.

**Inter-Class Separation in Feature Space:** To further analyze the structure of learned representations, we examine the distribution of inter-class distances in the penultimate feature space. fig. 2a shows a density plot comparing these distributions across different Active Learning strategies. Notably, NCAL-R exhibits a clear rightward shift, indicating larger average separation between class centroids (mean = 15.944), compared to Random (15.114), Coreset (15.070), and CDAL (15.130). This increased inter-class distance suggests that NCAL-R promotes more discriminative and geometrically separated class representations an essential property for improving generalization, especially under low-label regimes and OOD scenarios.

**Performance Comparison in Long-Tail Distribution:** Real-world data comes in a long-tail distributions, leading to bias towards certain classes. We construct a highly imbalanced version of ImageNet-100 by applying an exponential decay to class sample counts with a decay factor of $\beta = 0.05$, leading to a pool of 41,454 samples. An active-learning cycle with this pool achieves 45.15% for NCAL-R, compared to 42.30% (Random), 42.06% (Coreset) and 41.94% (CDAL) an improvement of +3% with only 16k images fig. 2b.

**Evaluating Transferability of ActiveOOD Strategies:** In this ablation, we evaluate the recently proposed ActiveOOD technique SISOMe (8) for Open-Set in our Closed-Set AL setup by removing its OOD filtering component. As shown in fig. 2c, SISOMe performs significantly worse than both standard baselines and NCAL. These results indicate that SISOMe's scoring heuristics do not transfer well to settings without explicit OOD filtering.

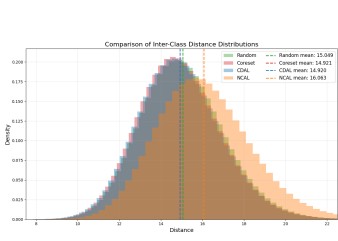
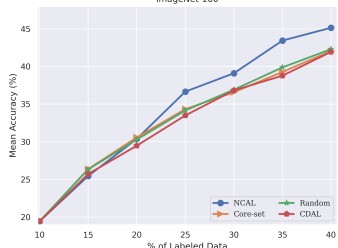
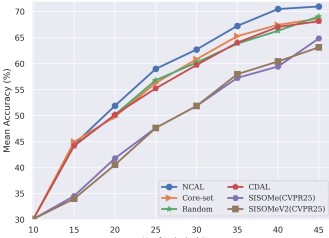

(a) Inter-Class Separation in Feature Space

(b) Comparison in Long-Tail Distribution

(c) Comparison with ActiveOOD

Figure 2: Ablation

## 4 Conclusion

We presented `NCAL-R`, an Active Learning framework that leverages Neural Collapse geometry. By combining CMAP and FF scores, NCAL selects structurally informative and uncertain samples, yielding more discriminative and robust feature spaces. Experiments show consistent gains across accuracy, OOD detection, OOD generalization, and category discovery. At its core, `NCAL-R` shows that structure matters – aligning acquisition decisions with the emergent geometry of deep networks can pay significant dividends.

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

## A Deriving the CMAP

In this section, we derive the Class-mean Alignment Perturbation (CMAP) score ($\delta_x$), as introduced in Sec. 2. The CMAP quantifies the change in alignment of the normalized class means induced by candidate sample $x$.

Let $x$ be a candidate sample with penultimate-layer feature embedding $z$, and let $c = f(x)$ be its predicted class. Denote $C$ as the number of classes. Suppose the current class mean for class $c$ is $\mu_c$, computed over $n_c$ training samples. If $x$ is added to class $c$, the updated class mean becomes:

$$\tilde{\mu}_c = \frac{n_c \mu_c + z}{n_c + 1}$$

Let $\mu_1, \ldots, \mu_k$ be the class means before adding $z$, and $\bar{\mu}_i = \mu_i/\|\mu_i\|$ be their $\ell_2$-normalized versions. Define the sum of all normalized class means: $\bar{M} := \sum_{i=1}^{C} \bar{\mu}_i$

Let $CMA_{\text{init}}$ and $CMA_{\text{final}}$ denote the Class-mean alignment (CMA) before and after adding the sample, respectively. This expression is the pair-wise average cosine similarity of class means. Then,

$$CMA_{init} = \frac{1}{k(k-1)} \sum_{\substack{i,j=1 \\ i \neq j}}^{k} \text{Sim}(\mu_i, \mu_j)$$

$$= \frac{1}{k(k-1)} \left[ \sum_{\substack{i,j=1 \\ i \neq j \\ i \neq c \\ j \neq c}} \text{Sim}(\mu_i, \mu_j) + 2 \sum_{\substack{i=1 \\ i \neq c}}^{k} \text{Sim}(\mu_i, \mu_c) \right]$$

We isolate the terms involving class $c$ since only those are affected by the perturbation. The remaining terms cancel when computing the delta:

$$\delta_x = CMA_{\text{final}} - CMA_{\text{init}}$$

$$= \frac{2}{k(k-1)} \sum_{\substack{i=1 \\ i \neq c}}^{k} \left[ \text{Sim}(\tilde{\mu}_c, \mu_i) - \text{Sim}(\mu_c, \mu_i) \right]$$

Using cosine similarity, $\text{Sim}(a, b) = \frac{a^T b}{\|a\|\|b\|}$, and denoting $\bar{\mu} = \frac{\mu}{\|\mu\|}$ as the unit-norm version of a vector, we simplify the expression:

$$\delta_x = \frac{2}{k(k-1)} \sum_{\substack{i=1 \\ i \neq c}}^{k} \left[ \bar{\tilde{\mu}}_c^T \bar{\mu}_i - \bar{\mu}_c^T \bar{\mu}_i \right]$$

$$= \frac{2}{k(k-1)} \sum_{\substack{i=1 \\ i \neq c}}^{k} \left[ (\bar{\tilde{\mu}}_c - \bar{\mu}_c)^T \bar{\mu}_i \right]$$

$$= \frac{2}{k(k-1)} (\bar{\tilde{\mu}}_c - \bar{\mu}_c)^T \sum_{\substack{i=1 \\ i \neq c}}^{k} \bar{\mu}_i$$

$$= \frac{2}{k(k-1)} (\bar{\tilde{\mu}}_c - \bar{\mu}_c)^T (\bar{M} - \bar{\mu}_c)$$

Finally, omitting the constant for interpretability and ranking purposes, we define the perturbation score:

$$\boxed{\text{CMAP}(x) := \delta_x = (\bar{\tilde{\mu}}_c - \bar{\mu}_c)^T (\bar{M} - \bar{\mu}_c)}$$

**Implementation note:** CMAP requires only the current per-class counts $\{n_c\}$ and means $\{\mu_c\}$ plus the feature $z$ for $x$; the increment $\tilde{\mu}_c$ can be computed cheaply and $\bar{M}$ updated incrementally if desired.

## B  Training Protocol

At each AL cycle:

1. Train $f_\theta$ on $\mathcal{L}$ until the Neural Collapse phase.
2. Compute CMAP and FF for all $x \in \mathcal{U}$.
3. Select $\mathcal{B}$ using the combined score, query labels, and update $\mathcal{L} \leftarrow \mathcal{L} \cup \mathcal{B}$.
4. Repeat until budget is exhausted.

**Experimental settings.**

1. **ImageNet100:** Initial pool consists of 10% randomly sampled data, i.e. 13,000 samples. In each iteration, we select 5% (i.e., 6,500) samples to be annotated and added to the pool for next iteration of training. We terminate the loop when our labelled pool reaches 40% of the training set.

2. **CIFAR100:** The initial pool size is 10%, i.e. 5,000 images, acquiring 5% (2,500 images) in each cycle. We terminate at 45% pool size.

3. **CIFAR10:** The initial pool is 2% (i.e. 1,000 images), acquiring 2% images every cycle until 20% pool size.

**Compute.** We run all our experiments on an A100 GPU with a 20 GB memory capacity.

# C  Algorithm Pseudo Code

1: **Input:** Unlabeled pool $\mathcal{U}$, labeled set $\mathcal{L}$, class means $\{\mu_c\}$, model checkpoints $\{f_t\}_{t=T_i}^{T_f}$, acquisition budget $k$
2: **Output:** Selected sample indices $\mathcal{A} \subset \mathcal{U}, |\mathcal{A}| = k$
3: Initialize empty lists $\{\delta_x\}$ and $\{\phi_x\}$ for each $x \in \mathcal{U}$
4: Compute normalized class means $\bar{\mu}_c := \mu_c / \|\mu_c\|$ for each class $c$
5: Compute $\bar{M} := \sum_c \bar{\mu}_c$
6: **for all** $x \in \mathcal{U}$ **do**
7:    $c \leftarrow f(x)$ {Predicted label for $x$}
8:    $z \leftarrow$ penultimate-layer feature of $x$
9:    $\tilde{\mu}_c \leftarrow \frac{n_c \mu_c + z}{n_c + 1}$
10:    $\bar{\tilde{\mu}}_c \leftarrow \tilde{\mu}_c / \|\tilde{\mu}_c\|$
11:    $\delta_x \leftarrow (\bar{\tilde{\mu}}_c - \bar{\mu}_c)^T (\bar{M} - \bar{\mu}_c)$
12:    $\phi_x \leftarrow 0$
13:    **for** $t = T_i + 1$ **to** $T_f$ **do**
14:      **if** $f_t(x) \neq f_{t-1}(x)$ **then**
15:        $\phi_x \leftarrow \phi_x + 1$
16:      **end if**
17:    **end for**
18: **end for**
19: Standardize scores using Z-score normalization:

$$\text{CMAP}(x) \leftarrow \frac{\delta_x - \mu_\delta}{\sigma_\delta}, \quad \text{FF}(x) \leftarrow \frac{\phi_x - \mu_\phi}{\sigma_\phi}$$

20: Compute acquisition scores: $s_x := \frac{\text{CMAP}(x) + \text{FF}(x)}{2}$ for each $x \in \mathcal{U}$
21: Select top-$k$ samples: $\mathcal{A} \leftarrow \text{TopK}(\{s_x\}_{x \in \mathcal{U}}, k)$
22: **return** $\mathcal{A}$

**Algorithm 1:** NCAL Acquisition Function

# D  Limitations of NCAL-R

**Limitations.** NCAL-R relies on models being trained into the *neural collapse* regime, i.e., the terminal phase where training accuracy plateaus and geometric regularities emerge. Reaching this phase can require many epochs, depending on the dataset and architecture, which may limit efficiency. Moreover, the study of Neural Collapse in large-scale models (e.g., LLMs) remains limited. Since such models are typically trained for only a few epochs, it is unclear whether NCAL-R's assumptions hold in these settings. We have not evaluated NCAL-R under such large-scale regimes, and adapting it there may require further investigation.

