# OpenReview forum: "Reliable Active Learning from Unreliable Labels via Neural Collapse Geometry"
_NeurIPS.cc/2025/Workshop/Reliable_ML — NeurIPS 2025 - Reliable ML Workshop_

### Official Review · Reviewer_RwRf · 2025-09-19
**Needs improvement**

**Rating:** 5
**Confidence:** 2

**Review:**

Summary: This paper studies active learning when the training model has entered the Neural Collapse regime. The motivation is that active learning may not be reliable with noisy labels or data distribution shifts. This paper proposes two scores for each datapoint, indicating whether the data would stabilize/distort inter-class geometry, and temporal instability of representations across training checkpoints. The pipeline then selects data with the highest score for active learning. Experiments on ImageNet-100 and CIFAR-100 show that this method achieves high accuracy with fewer samples, improved robustness against noise, and better generalization.

Comment: I am not an expert on active learning nor neural collapse theory. I think it would be helpful to include more background knowledge on Neural Collapse and a more complete literature review on existing methods for active learning. A discussion on the extent to which the Neural Collapse hypothesis holds in modern deep learning would be helpful. The two scores seem heuristically reasonable but lack a theoretical justification for why they improve label efficiency and robustness to distribution shift. Also, the computational efficiency of the pipeline is not discussed or compared with other baselines. Overall, I think this paper offers some interesting observations in active learning using the Neural Collapse assumption but the limited theoretical and empirical evaluation make it not convincing enough to support the claims.

---

### Official Review · Reviewer_kmg5 · 2025-09-22
**interesting preliminary results**

**Rating:** 7
**Confidence:** 3

**Review:**

1. This paper proposes a new active learning method that prioritizes new points to sample based on not just regions of the domain with high overall uncertainty but also higher likelihood to affect internal representations of class boundaries.
2. The paper is overall easy to follow. [Note this is not my area so I cannot comment on novelty.] Reasonable and promising approach.
3. It's not clear to me whether there is really a huge separation across the methods (and whether CIFAR/MNIST are sufficiently challenging for a larger separation to emerge), but this is a workshop and initial results seem promising. I'm not sure what the numbers in Table 2 mean. It would be nice (for a method that has principled motivations) to have theoretical results as a complement.